# Roles of Extracellular Vesicles in Multiple Sclerosis: From Pathogenesis to Potential Tools as Biomarkers and Therapeutics

Cristiana Pistono [1],*, Cecilia Osera [1], Mariaclara Cuccia [1] and Roberto Bergamaschi [2]

1   Laboratory of Immunogenetics, Department of Biology & Biotechnology "L. Spallanzani", University of Pavia, 27100 Pavia, Italy; cecilia.osera@unipv.it (C.O.); mariaclara.cuccia@unipv.it (M.C.)
2   Inter-Department Multiple Sclerosis Research Centre, National Neurological Institute "C. Mondino", 27100 Pavia, Italy; roberto.bergamaschi@mondino.it
*   Correspondence: pistono.cristiana@gmail.com

**Abstract:** Extracellular vesicles (EVs) are involved in the regulation of immune system functioning and central nervous system (CNS) homeostasis, suggesting a possible role in multiple sclerosis (MS). Indeed, by carrying several types of mediators, such as cytokines, enzymes, and RNAs, EVs can display both anti- and pro-inflammatory roles on the innate and adaptive immune system, and are involved in several CNS functions, including neuronal plasticity, trophic support, disposal of cellular components, axonal maintenance and neuroprotection. In this review, we provide an overview of the studies carried out to understand the role of EVs in the compromised immune system and CNS functioning typical of MS. Moreover, we also highlight the potential of EVs for the diagnosis of this disorder, thanks to their ability to cross the blood-brain barrier (BBB). In addition, we describe the advances in the use of EVs as therapeutic agents by describing their therapeutic potential.

**Keywords:** extracellular vesicle; multiple sclerosis; biomarker; therapy

## 1. Introduction

Multiple sclerosis (MS) is a chronic inflammatory disorder involving the central nervous system CNS, characterized by early onset, usually between 20 and 40 years of age, that represents the most common cause of neurologic disability in young adults [1]. The etiology is complex and not fully elucidated, with genetic susceptibility and environmental and epigenetic factors playing a key role in disease development [2,3]. MS pathogenesis involves different mechanisms, including inflammatory phenomena, blood–brain barrier (BBB) breakdown, demyelination, and neurodegeneration, and it is characterized by the formation of plaques localized both in the white and in the grey matter of the CNS [4]. The typical demyelinated MS lesions are characterized by foci of inflammation, due to the infiltration of peripheral immune system cells, such as T and B lymphocytes and monocytes, and are surrounded by reactive glial cells [5].

In recent decades, a growing number of studies highlighted the role of extracellular vesicles (EVs) in the pathogenesis of both immune and neurodegenerative disorders [6,7]. Potentially all types of cells can release EVs, double membrane-enclosed small structures that are released in the extracellular space and carry a plethora of molecules, including lipids, DNA, RNA, and proteins. EVs can be found in all body fluids and act as mediators of intercellular communication [8]. Conventionally, EVs are often divided into two main classes, based on their size, origin, and biogenesis: exosomes originate from the endosomal pathway and are characterized by a size between 30 and 150 nm, whereas microvesicles have a size between 100 and 1000 nm and are shed through outward budding of the plasma membrane [9]. However, a precise classification remains challenging, and the MISEV2018 guidelines promote the use of the umbrella term "EV" [10]. The EVs' content varies based on the type of cell from where they are shredded, the physiological state of the cells, and the environmental conditions [8]. EVs have been described to be involved in a plethora of

processes, including immune response and response to stress [11,12]. EVs released from both immune and non-immune cells can have a role in immune regulation; for instance, they can promote antigen presentation, thus stimulating the adaptive immune response, and, on the other hand, they can mediate immunosuppressive functions [13]. EVs have also a role in CNS homeostasis: they are, for example, involved in trophic support of neurons and maintenance of myelin and synaptic plasticity [14].

EVs are thus involved in the regulation of both the immune system and CNS homeostasis, both related to the pathogenesis of MS. In addition, EVs can cross the (BBB) [15], suggesting a possible role in the compromised communication between the immune system and CNS, a feature that is typical of MS. In addition, the ability of EVs to cross the BBB makes them a potential tool for early diagnosis of neurological disease [16] and potential vehicles for targeted and non-invasive therapies [17].

The research on the role of EVs in neurodegenerative disease and MS is still progressing, together with the effort to take advantage of the potential role of EVs in treating MS. In this review, we make on overview of the advances of EVs' role in the pathological mechanisms involving the immune system and the CNS in MS, and we focus on the potential of EVs for the diagnosis and treatment of this disorder.

## 2. EVs in Immune Response and BBB Permeability

EVs released by immune and non-immune cells can have a pro-inflammatory role on innate immune cells, thanks to their ability to transfer several types of mediators, such as acute-phase proteins, and cytokines, enzymes, and RNAs that have an impact on activation, differentiation, and recruitment of innate immune cells, and cytokine production [18–20]. On the other hand, the anti-inflammatory effects of EVs have been described too; for instance, endothelial cell-derived EVs, carrying heat shock protein family A member 12B (HSPA12B), can downregulate the activation of nuclear factor-κB (NF-κB) in macrophages [19]. Furthermore, EVs have a role in adaptive immunity; they are involved in the process of lymphocyte development. EVs released by thymic epithelial cells carry tissue-restricted antigens to thymic conventional dendritic cells (cDCs) for antigen presentation, thus contributing to the negative selection of T-cells [21]. Moreover, thymic epithelial cell-derived EVs have a role in inducing the maturation of single-positive (CD4$^+$ or CD8$^+$) cells [22].

Like other cells, all immune cell types secrete EVs which play a fundamental role in immune innate and adaptive immunity [13]. EVs secreted by dendric cells (DCs) at different stages of activation are also able to target T-cells. In particular, EVs released by DCs can induce T-cell activation and expansion, and can have both Th1 and Th2 polarizing capacities, depending on the different EV subtypes released [23]. EVs derived from DCs carry cell surface molecules like major histocompatibility complex (MHC), intercellular adhesion molecule 1 (ICAM-1), and some other costimulatory molecules which can promote T-cell activation [24]. In turn, EVs secreted by T-cells can modulate the function of other cells, including DCs. T regulatory cells (Tregs) secrete EVs, which may promote a tolerogenic phenotype of DCs by increasing interleukin-10 (IL-10) and decreasing IL-6 production, thanks to the transport of miRNAs such as miR-150-5p and miR-142-3p [25]. Hence, functions of immune cell derived-EVs might depend on the types of cellular events and the physiological environment. For example, they may have a role in the germline center reaction by promoting survival and proliferation of B-cells and antibody switching class: EVs are required for the germinal center reaction and antibody production in vivo, thanks to the transfer of microRNAs mediated by EVs from T-cells–cell origin [26]. In addition to microRNA, proteins like PKM2 (pyruvate kinase muscle isozyme 2) may be involved in the T-cell-regulated B-cell antibody production [27]. EVs from activated T-cells contain key metabolic, as well as signaling components that can also target mast cells and other resting T-cells [28]. T-cell-derived EVs target mast cells, which can induce MAPK, thus promoting the release of IL-8 and oncostatin [29]. Treg cells release EVs that have a role in immunosuppressive activities [30]. EVs can participate in antigen presentation; peptide-

MHC (pMHC)-carrying EVs can attach to the surface of DCs, thus increasing the efficacy of antigen presentation [31]. In addition, vesicles carrying pMHC or the intact antigen can be internalized by antigen- presenting cells, contributing indirectly and after their processing, to antigen presentation [32].

In MS, the role of immune cells is central, triggering neuroinflammation by producing inflammatory cytokines and damaging myelin and oligodendrocytes [33] (Figure 1). EVs have been shown to have a role in autoimmune diseases, such as systemic lupus erythematosus and rheumatoid arthritis [6,34]. In MS, EVs seem to act as inflammatory mediators, as shown by several experiments. Willis and colleagues suggested an association between plasma-derived EVs and CD8$^+$ T-cells in the experimental autoimmune encephalomyelitis (EAE) mouse models of MS. The administration of EVs from the plasma of naïve C57BL/6 mice to MOG$_{35-55}$ EAE mice at the peak of the clinical disease led to the development of a spontaneous phenotype of relapsing-remitting disease, a feature not generally observed in this such mouse model [35]. These plasma-derived pEVs contain fibrinogen, as also observed in the plasma-derived EVs of relapsing-remitting multiple sclerosis (RRMS) patients, which induce robust CD8$^+$ T-cell response, whereas usually MOG$_{35-55}$ EAE induces a prominent CD4$^+$ T-cell-driven disease [36], and it is consistent with the observation that CD8$^+$ T-cells are the predominant T-cell subtype found within active lesions in RRMS patients [37]. EVs alteration has been observed in MS patients: in the plasma of RRMS patients in relapse, erythrocyte-derived EVs were increased, while platelet-derived (CD41b), leukocyte-derived (CD45), and CD4$^+$ T-cell-derived EVs were decreased compared to healthy controls. Endothelium-derived EVs were increased in stable RRMS patients compared to healthy controls [38]. Moreover, the cargo of EVs results altered in MS: T-cells from RRMS patients cultured in vitro were shown to release EVs with higher levels of miR-326 EVs, compared to healthy controls [39]. This microRNA is thought to play a role in MS immunopathogenesis by inducing Th17 cell differentiation and maturation [40]. In vitro, EVs purified from the plasma of MS patients can inhibit the induction of human IFN-γ-IL-17-Foxp3$^+$CD4$^+$ T-cells, considered as most suppressive Treg cells. The miRNA let-7i was markedly increased in the EVs from patients and can inhibit the differentiation of naive CD4$^+$ T-cells into functional Treg cells in vivo Treg cells [41]. EVs released by B-cells of patients with RRMS cultured in vitro are enriched with proteins, which have the potential to induce oligodendrocyte death, but did not induce toxicity either in astroglia or microglia [42].

EVs from several immune cells seem to be significatively increased in untreated RRMS patients with low disability, despite very limited changes in circulating immune cells [43]. EVs purified from the serum of patients with MS express higher levels of Epstein-Barr virus (EBV) proteins, like EBV nuclear antigen EBNA1 and latent membrane proteins LMP1 and 2A, compared to healthy subjects. The in vitro incubation with EBV-positive EVs induced C-X-C motif chemokine ligand 10 (CXCL10) and C-C motif chemokine ligand 2 (CCL2) secretion by monocyte-derived macrophages. Notably, monocyte-derived macrophages differentiated from patients with active MS were better secretors of CXCL10 and other interferon-γ-inducible chemokines, like CCL2 and C-X-C motif chemokine ligand 9 (CXCL9), compared to the cells from healthy subjects and patients with a stable for of the disease [44].

The pro-inflammatory environment typical of MS, established by the activation of immune cells, induces the endothelial cells to secrete cytokines and metalloproteinases, thus contributing to the disruption of the extracellular matrix and endothelial tight junctions of the BBB. This leads to the infiltration of immune cells, including lymphocytes, into the CNS and contributes to the dysfunction of the BBB, a characteristic of MS pathogenesis [45]. EVs seem to be also involved in this process: EVs released by T-cells and containing chemokine CCL5, and arachidonic acid can increase the expression of ICAM-1 on endothelial cells, and of Mac-1 on monocytes [46–48]. On the other hand, endothelial cell-derived EVs are able to activate CD4$^+$ and CD8$^+$ cells: such EVs isolated from the serum of healthy subjects can induce the expression of molecules like MHC class II, CD40, and inducible T-cell co-

stimulator ligand [49]. In the context of MS, it has been observed that EVs from the serum of active RRMS patients can enhance the trans-endothelial migration of monocytes through a monolayer in vitro model of BBB [50]. Unstimulated cultured monocytes isolated from RRMS patients release higher levels of EVs compared to cells isolated from healthy controls, but upon stimulation with benzoyl-ATP (BzATP), only the monocytes from healthy controls showed a significant increase in EVs production, as if the monocytes from MS patients were already per se stimulated [51]. Interestingly, interferon-beta and teriflunomide, two medications used for MS, may reduce the production of EVs from patients' monocytes in a time-dependent manner, but only teriflunomide leads to the downregulation of purinergic P2X7 receptor and inflammasome components expression [51]. CNS endothelial-derived EV can be found in blood samples, and their level could be an index of BBB permeability and integrity [52].

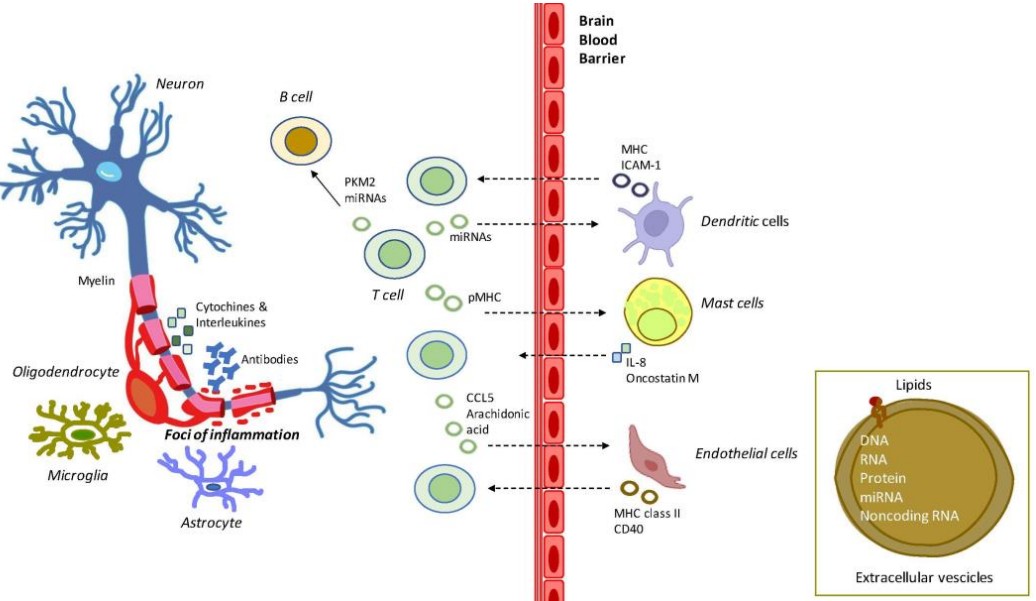

**Figure 1.** Exemplification of the role of EVs in MS inflammation and demyelination. MS is characterized by the activation of brain microglial and astrocyte cells and by a massive infiltration of T- and B-cells. T-cells release cytokines and chemokines that damage the oligodendrocytes, and the myelin sheath. B-cells produce myelin-specific antibodies, thus further damaging myelin sheath at *foci* of inflammation. In this context, EVs produced by T-cells are released to reach out to mast cells and endothelial cells, and dendritic cells, which further contributes to MS pathology. The EVs flux is supposed to be bidirectional across the BBB. Abbreviations: CCL5: C-C motif chemokine ligand 5; CD40: cluster of differentiation 40; MHC: major histocompatibility complex; ICAM-1: intercellular adhesion molecule 1; IL-8: interleukin 8; miRNAs: microRNAs; PKM2: pyruvate kinase muscle isozyme 2; pMHC: peptide-MHC.

## 3. EVs in the CNS

EVs are chairman in the intricate communication system between neurons and glia, which allows the correct functioning of the CNS. EVs in the nervous system are involved in several functions, including neuronal plasticity, trophic support, disposal of cellular components, axonal maintenance, and neuroprotection. In addition, they can deliver signals across CNS barriers [14] (Figure 2). EVs secreted by glial cells have neuroprotective and homeostatic effects, but several works also propose a detrimental effect in pathological conditions [53].

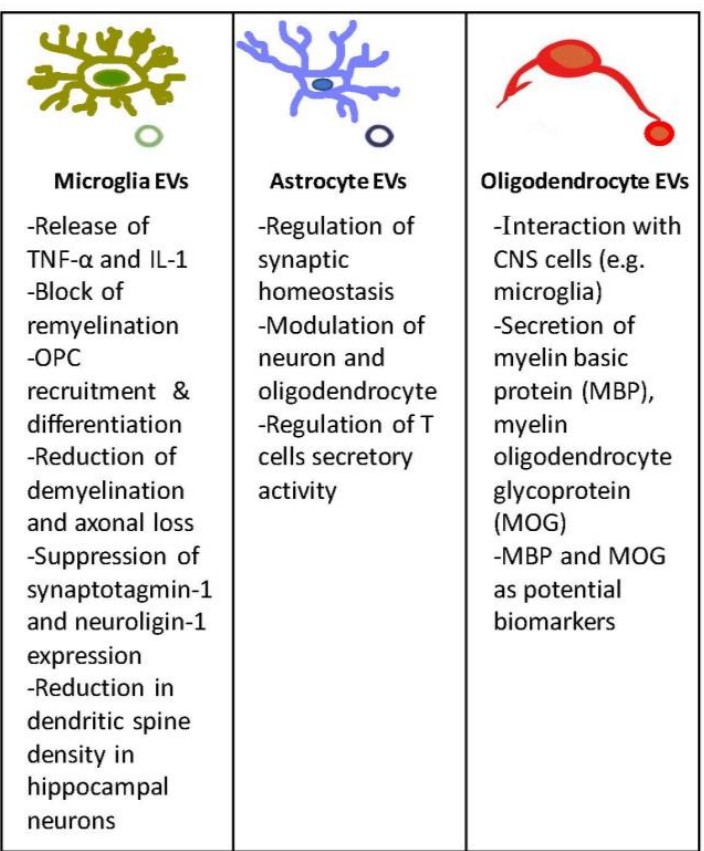

**Figure 2.** EVs in the CNS. In this figure are depicted the main features and functions of EVs released by microglia, astrocyte, and oligodendrocyte cells.

Microglia constantly monitor and respond to changes in the CNS environment, representing the resident immune cells of the CNS: microglia detect the presence of pathogens and tissue injury, contribute to the control of neurotransmission, and, thanks to the bidirectional neuronal-microglial communication, contribute to sculpture the neuronal connections during development and, later, for the maintenance of CNS homeostasis and brain plasticity [54,55]. Not only chemokines and cytokines are important for microglia-neuron communication, but increasing evidence supports the central role of EVs in the control of neuronal morphology and synaptic plasticity, and in the neuroinflammatory processes [56,57]. Indeed, it has been shown that EVs from activated microglia carry pro-inflammatory mediators, such as TNF-$\alpha$ and IL-1, and have a distinct proteomic profile [58,59], potentially contributing to the propagation of inflammatory stimuli throughout the CNS and enforce inflammation in neuroinflammatory diseases [60]. Furthermore, microglia influence remyelination, with a predominant pro-regenerative phenotype essential for efficient myelin repair after damage [61]. Although this is well-known, how microglia can have an impact on myelin repair was unclear, but a 2019 study described that EVs released by pro-inflammatory microglia could block remyelination, contrary to the EVs released from pro-regenerative microglia, which are able to promote recruitment and differentiation of oligodendrocyte precursor cells (OPCs) [62]. Interestingly, in vitro experiments revealed that the blockade of OPC maturation only occurred when microglia were co-cultured with astrocytes, implicating these cells in remyelination failure [62]. This fact remarks that demyelinating lesions may fail to remyelinate because of EVs released by chronically activated microglia, which can halt OPC differentiation by inducing harmful astrocyte conversion and canceling the direct pro-myelinating proprieties of EVs.

In MS, the involvement of microglia-EVs has been analyzed in several studies. In the mouse model of EAE, the concentration of EVs released from microglia/macrophages is significantly increased compared to naïve mice, and seems to be associated with disease

course, peaking at onset and during clinical relapses, and dropping in the chronic phase of the disorder [63]. Interestingly, the same study highlighted the presence of higher levels of myeloid EVs in the cerebrospinal fluid (CSF) of MS patients. In addition, the injection of EVs released from cultured microglia into the corpus callosum of EAE mice fosters the infiltration of CD45$^+$ cells and promotes the recruitment of Iba-1+ ameboid cells at the injection site, compared to mice injected with liposomes, suggesting that EVs may contribute to the development of focal inflammatory lesions in EAE [63]. The fact that these EVs may have a pathogenic role in EAE is also suggested from the observation that knock-out mice for Acid SMase, an enzyme of the sphingomyelinase's family essential for EVs shedding from glial cells [64], were protected from EAE [63]. In a 2018 study, the authors engineered BV-2 cells to release EVs overexpressing the endogenous signal Lactadherin (Mfg-e8), which targets phagocytes, and carries the anti-inflammatory cytokine IL-4. A single injection of these EVs into the cisterna magna of EAE mice at the onset of the disease resulted in a significant decrease in clinical signs by a reduction of demyelination and axonal loss [65]. Although the majority of the works about microglia-derived EVs in MS are mainly focused on their ability to mediate inflammation and contribute to the propagation of neuroinflammation thought the CNS, it is important to consider that microglia can contribute to the regulation of synaptic function and plasticity, thus playing a role in neuronal network formation and information processing [66]. Indeed, synaptic plasticity is deeply altered in MS [67,68], and it has been shown that EVs from microglia can either stimulate synaptic activity [69] or inhibit presynaptic transmission by activating presynaptic cannabinoid receptor type 1 expressed on GABAergic neurons [70]. Prada and colleagues showed that microglia-derived EVs transfer miR-146a-5p to neurons suppressing synaptotagmin-1 and neuroligin-1 expression [71]. Morphological analysis showed that prolonged exposure to inflammatory EVs led to a reduction in dendritic spine density in hippocampal neurons both in vivo and in vitro. Furthermore, the dendritic spine loss is accompanied by a decrease in the density and strength of excitatory synapses, thus finding a link between inflammatory microglia, increased EV production, and loss of excitatory synapses. Interestingly, in the same work, miR-146a-5p was found in the EVs isolated from CSF of MS patients, where EVs of myeloid origin account for ~65% of total EVs detectable, suggesting a possible link between microglia activation and cognitive symptoms in MS [71]. However, studies on the role of microglia-derived EVs in MS synaptopathy and cognitive symptoms are still lacking, thus opening a new research topic that deserves future investigation.

Astrocytes are the most abundant cell type in the CNS and have a central role in its homeostasis by regulating several processes, including BBB permeability, nutrient uptake, and removal of metabolic waste. Furthermore, astrocytes protect neurons from neurotoxicity and cell death, and have a role in regulating neurogenesis and synaptogenesis [72,73]. In the communication among astrocytes and neurons, EVs play a central role: astrocytes can release EVs enriched with miRNAs that regulate synaptic homeostasis [74]. Moreover, astrocytes act as immune-competent cells, able to respond to and activate immune responses by secreting cytokines and chemokines. Interestingly, the in vitro treatment of astrocytes with TNF-$\alpha$ promotes the release of EVs, highlighting the EV release in inflammatory conditions [75].

It is known that astrocytes play multiple, even opposing, roles in MS, tuned by several factors, such as the nature and the severity of the CNS insults, their localization, the local pro- or anti-inflammatory environment, and the crosstalk with CNS and infiltrated immune cells. Indeed, they can contribute to the recruitment and retention of leukocytes at the lesion sites and to the establishment of an inflammatory loop that mediates the disease. Moreover, these cells play a role in tissue damage through neurotoxic activities, thus promoting neurodegeneration and disease progression. On the other hand, astrocytes may also restrict inflammation and promote neuroprotection [76]. However, few studies about the EVs released from astrocytes are described in the literature. Ulivieri and colleagues focused their attention on the deficiency of Rai, a member of the Shc family of protein adaptors,

which are cytoplasmic signal transducers also expressed in astrocytes [77]. EAE-deficient mice are protected from demyelination and reactive astrogliosis; indeed, Rai contributes to the generation of a pro-inflammatory microenvironment in the CNS [78]. The same research group analyzed the impact of Rai expression on astrocyte function under basal conditions and in response to IL-17 treatment: Rai contributes to neurotoxic pathways and, more interestingly, Rai may modulate the protein contents of astrocyte-derived EVs, thus modulating neuron and oligodendrocyte function in MS [79]. In addition, a more recent article suggested the role of astrocyte-derived EVs in the regulation of T-cells' secretory activity. Indeed, the authors reported that in vitro EV from human primary astrocytes with various phenotypes differently affects, in a different way, the production of IFN-γ, IL-17A, and CCL2 by CD4$^+$ T-cells from MS patients and healthy subjects [80].

Oligodendrocytes, highly specialized cells responsible for myelin production and targets of inflammatory and immune attacks typical of MS, are somewhat interesting [81]. A growing body of evidence suggests oligodendrocyte lineage might have a more central role in the origin and progression of MS [82], thanks to the ability of these cells to express immunomodulatory factors [83]. Like the other glial cell types, oligodendrocytes secrete EVs which are able to interact with other CNS cells [84,85]. Studies about oligodendrocyte-derived EVs are still few and mainly focused on their potential use as biomarkers; Agliardi and colleagues, which focused on myelin basic protein (MBP) and myelin oligodendrocyte glycoprotein (MOG) content in oligodendrocytes-derived EVs from the serum, suggested their use as biomarkers to help in the diagnosis of clinical MS phenotypes [86].

## 4. EVs as Biomarkers

Biomarkers can be used as a diagnostic tool to identify patients with a disease in its early stages, assess the staging of the disorder, indicate the prognosis, and predict and monitor the clinical response to treatments [87]. In recent decades, the interest in circulating EVs as potential biomarkers in liquid biopsy has grown exponentially [88–90]. Indeed, EVs originating from a variety of cell sources are highly stable and can be found in all body fluids; their cargo can be analyzed and reflect the physiopathologic conditions of the parental cells [91]. The main biological fluids in which circulating EVs have been identified are plasma, serum, urine, saliva, breast milk, and CSF [92]. EVs are being studied as biomarkers for multiple neurodegenerative diseases, including MS.

### 4.1. Plasma/Serum

Blood, one of the easiest accessible sources of biomarkers, represents only a minimally invasive medical procedure for patients carrying EVs that can be isolated in a relatively easy way. Circulating EVs are a mixture of vesicles shredded from various cell types: EVs released from CNS cells can cross the BBB and reach the bloodstream [15], thus circulating EVs may represent an accessible source of CNS biomarkers, which can potentially reveal the pathological processes of the CNS during MS. Furthermore, the immune cells involved in the inflammatory response typical of MS also release EVs into the bloodstream [13] that can potentially give important information about the pathological inflammation typical of MS. When analyzing the circulating EVs, studies must specifically address the cell type of EV origin, directing the analysis towards EVs from the CNS and immune system cells to provide information about the pathologic processes in MS. EVs released from CNS cells represent a small part of the total circulating EV population, thus enrichment techniques, such as immunoprecipitation using antibodies targeting the neuronal marker proteins are a necessary step to detect CNS biomarkers for MS [93]. Among the used markers, there are L1 cell adhesion molecule (L1CAM) and NCAM to detect EVs of neuronal origin [16,94], glutamate transporter primarily expressed by astrocytes (GLAST) to isolate astrocyte-derived EVs [95], CD11b and IB4 to obtain microglia-derived EVs [96]. In addition, CD31 and CD51 are used to isolate EVs from endothelial cells [97,98], CCR5, and CCR3 from lymphocytes [99]. Among the different bioactive elements carried by plasma and serum EVs, studies mainly focused on their protein and nucleic acids (mainly microRNAs) content,

as reported in Table 1. Alterations in protein content are linked to MS presence and progression and focus on EVs from different cells type, like endothelial cells, involved in endothelial dysfunction [52,97,98,100], and cells present in the blood, such as platelets and leukocytes [101,102]. EVs from cells of the CNS, recovered from the blood, have the potential to become MS biomarkers [95]. In addition, changes in miRNA profiles in MS patients have been analyzed as potential markers to diagnose MS, predict disease subtype and clinical course [86,103,104], and monitor treatment response [105,106].

**Table 1.** Potential biomarkers from the analysis of EVs from serum/plasma of MS patients.

| Plasma/Serum | EV Origin | Surface Marker | Detection Level | Potential Application | Reference |
|---|---|---|---|---|---|
| Plasma | Endothelial cells | CD31 CD51 | ↑CD31+ EVs in MS in exacerbation vs. HC ↑CD51+ EVs in MS in exacerbation and remission vs. HC | Endothelial dysfunction | [97] |
| Plasma | Endothelial cells | CD31 CD62 | ↑CD54+ EVs in MS in exacerbation vs. MS in remission and HC ↑CD62+ in MS in exacerbation vs. MS in remission and HC | Endothelial dysfunction during exacerbation | [100] |
| Plasma | Platelets | CD62p | ↑CD62p+ EVs in MS vs. HC | Platelet activation and monocyte interaction with damaged endothelium | [102] |
| Plasma | Platelet Monocytes Leukocytes | CD61 CD14 CD45 | ↑CD61+ EVs in untreated MS vs. HC ↑CD61+, CD14+, CD45+ EVs in RRMS vs. HC and SPMS | Platelet activation, monocyte, and leukocyte interaction with damaged endothelium | [101] |
| Serum | Endothelial cells | CD31 CD51 CD61 CD54 | ↑CD31+/CD51+/CD61+/CD54+ EVs in RRMS vs. HC ↓CD31+/CD51−/CD61−/CD54− EVs in SPMS vs. HC | BBB dysfunction | [98] |
| Serum | | | ↑MOG EV content in RRMS patients in relapse and SPMS vs. HC | Disease activity | [107] |
| Serum | | | ↓TLR3 and ↑TLR4 in RRMS EVs vs. HC | Altered balance of innate immune signaling | [108] |
| Plasma | Neurons Astrocytes | L1CAM + NEVs GLAST + AEVs | ↓synaptopodin and synaptophysin in neuron-derived EVs in MS vs. HC ↑C1q, C3, C3b/iC3b, C5, C5a, factor H in astrocyte-derived EVs in MS vs. HC | Synaptic loss | [95] |
| Plasma | CNS endothelial cells | Absence of CD3 CD41 Presence of CD31, CD105 CD144 CD31 CD105 CD144 MAL | ↑concentration of CNS endothelial-derived EV in active vs. stable MS and HC | BBB permeability and active disease | [52] |
| Serum | Oligodendrocytes | MOG | ↑MBP levels in EVs in CIS, RRMS, and PPMS vs. HC ↑MBP levels in EVs in PPMS vs. RRMS and CIS MBP levels correlate with EDSS and MSSS | Diagnosis, prediction of disease subtype | [86] |
| Serum | | | ↓hsa-miR-122-5p, hsa-miR-196b-5p, hsa-miR-301a-3p, and hsa-miR-532-5p in MS patients with relapse vs. HC | RRMS activity | [109] |
| Serum | | | ↑miR-15b-5p, miR-451a, miR-30b-5p, miR-342-3p in RRMS vs. HC ↑miR-127-3p, miR-370-3p, miR-409-3p, miR-432-5p P/SPMS vs. HC | Diagnosis, prediction of disease subtype | [103] |

**Table 1.** *Cont.*

| Plasma/Serum | EV Origin | Surface Marker | Detection Level | Potential Application | Reference |
|---|---|---|---|---|---|
| Serum | | | ↓hsa-miR-486-5p, hsa-miR-451a, hsa-let-7b-5p, hsa-miR-320b, hsa-miR-122-5p, hsa-miR-215-5p, hsa-miR-320d, hsa-miR-19-3p, hsa-miR-26a-5p, hsa-miR-142-3p, hsa-miR-146a-5p, hsa-miR-15b-3p, hsa-miR-23a-3p, hsa-miR-223-3p in IFN-β treated MS patients vs. non treated patients ↑hsa-miR-22-3p, hsa-miR-660-5p in IFN-β treated MS patients vs. non-treated patients | Treatment monitoring | [106] |
| Plasma | | | ↑EVs after 5 h of treatment with fingolimod vs. pre-treatment | Treatment monitoring | [105] |
| Plasma | Myeloid cells | IB4 | ↑miR-150-5p and ↓let-7b-5p in cognitively impaired MS patients vs. cognitively preserved MS patients | Cognitive deficits in MS | [110] |
| Serum | | | ↓miR-4697-5p, miR-711, miR-4761-3p, miR-5094, miR-4474-5p, miR-1909-3p in CIS-remission vs. HC and RRMS-relapse ↑miR-4787-5p, miR-135b-5p, miR-5192, miR-451a, miR-6811-3p, miR-4476, miR-16-5p, miR-1909-3p, miR-6840-3p in RRMS-relapse vs. HC and CIS-remission. | Diagnosis, prediction of disease subtype | [104] |

Abbreviations. CIS: clinically isolated syndrome; CNS: the central nervous system; EVs: extracellular vesicles; HC: heathy controls; MBP: myelin basic protein; MOG: myelin oligodendrocyte glycoprotein; MS: multiple sclerosis; PPMS: primary progressive multiple sclerosis; RRMS: relapsing-remitting multiple sclerosis; SPMS: secondary progressive multiple sclerosis.

### 4.2. Cerebrospinal Fluid

Although the collection of CSF is an invasive procedure with a small volume obtained, the main advantage of CSF is that it accurately reflects the inflammatory profile of the CNS [111]. The CSF is a source of EVs originating directly from the CNS. Indeed, due to its proximity to the CNS, CSF is rich in EVs with neuronal, microglial/macrophagic, oligodendroglial, and astrocytic markers. Although isolating EVs from CSF is still challenging, a growing number of studies are present in the literature, with different protocols beings used. Several works analyze the levels of EVs to detect changes related to the presence of MS and different phases of the disorder [63,99,112], as shown in Table 2. Furthermore, proteomics studies have also been performed to check the EV content: quantitative proteomic analysis of EVs from CSF detected signature proteins differentiating neuromyelitis optica from MS, with 59 and 123 significantly altered proteins, respectively, which are involved in processes related to cellular adhesion, immune response, and cellular structure development [113]. Welton and colleagues identified 50 proteins significantly and exclusively enriched in CSF-EVs of RRMS patients, compared to non-demyelinating controls, revealing a strong association between these proteins and biological processes relating to MS pathology [114].

A 2019 work considered the EVs present in tears, which are related to the CNS, and found microglia-derived and neural-derived EVs present in this body fluid. The EV analysis revealed that EVs from both CSF and tears of MS patients carry similar proteins involved in inflammation, angiogenesis, and immune response, demonstrating an EVs-mediated molecular cross-talk between CSF and tears [115]. This study may pave the way to new diagnostic perspectives because tears are highly informative and easily accessible, thus representing one of the most convenient body fluids for biomarker applications.

**Table 2.** Potential biomarkers from the analysis of EVs from CSF of MS patients.

| EV Origin | Surface Marker | Detection Level | Potential Application | Reference |
|---|---|---|---|---|
| Myeloid cells | IB4 | ↑EVs in RRMS and CIS vs. HC | Microglia/macrophage activation | [63] |
| Myeloid cells Subset of CD8 memory T-cells Th2 cells Th1 cells | IB4 CCR3 and CCR5 CD4 and CCR3 CD4 and CCR5 | ↑EVs in clinical relapse vs. remission ↑EVs in clinical relapse vs. remission ↑CCR3+/CCR5+ EVs in patients with gadolinium-enhanced MR lesions ↑CD4$^+$/CCR3+ EVs in patients with gadolinium-enhanced MR lesions ↑CD4$^+$/CCR5+ EVs in patients with gadolinium-enhanced MR lesions | Identification of different MS phases | [99] |
| Myeloid cells | IB4 | ↑EVs in CIS patients vs. HC | Risk stratification | [112] |

Abbreviations. CIS: clinically isolated syndrome; EVs: extracellular vesicles; HC: heathy controls; RRMS: relapsing-remitting multiple sclerosis.

### 4.3. Clinical Applications of Potential EV-Derived Biomarkers

Based on the literature data, EVs from nervous, immune, platelet, and endothelial cells may be a source of biomarkers to differentiate MS subtypes, provide insight into disease activity and progression, and information about the response to treatments. However, the biomarkers detected in those studies still need confirmation in larger cohorts. In fact, the clinical experimental evidence of EVs as potential biomarkers for MS is still in its early stages. Conflicting results, heterogeneity, and lack of replication are ongoing challenges. The problem of small sample size, biospecimen collection methods, and profiling techniques can be overcome with appropriate planning and protocol standardization, which may allow the comparison of the results of different studies. Indeed, biomarker validation is crucial for the use of EVs for diagnosis in a clinical setting. A promising field is the one of miRNA profiling from EVs isolated from plasma or serum. Some microRNAs were found to be associated with MS in more than one study. For example, a reduction of miR-122-5p was observed in EVs isolated from the serum of MS patients with relapse, compared with healthy subjects [109]. However, the reduction was also described in IFN-β treated MS patients, compared to non-treated patients [106]. A major group of transcripts regulated by miR-122-5p are transcriptional and DNA-interacting factors, including signal transducer and activator of transcription 3 (STAT3) and cell cycle regulator aryl hydrocarbon receptor (AHR) 43 transcripts [109]. These proteins are regulators of Th17 cells and Treg differentiation [116,117]. More studies are necessary to understand the role of miR-122-5p in MS, since the dysregulation of several microRNAs may impact the alteration of the differentiation of Th17 and Treg cells. miR-451a levels were increased in EVs from the plasma of RRMS patients, compared to heathy controls, and in EVs from the serum of RRMS patients during relapse, compared with heathy subjects and patients with CIS during remission [103,104]. Furthermore, miR-451a levels were reduced in EVs from the plasma of IFN-β treated MS patients, compared to non-treated patients [106]. miR-451a was described as a regulator of oxidative stress [118] with a potential impact on several neurodegenerative processes [119]. Another miRNA, hsa-let-7b-5p, was reduced in EVs from the serum of IFN-β treated MS patients, compared to non-treated ones [106]. The same microRNA was found to be reduced in myeloid EVs from the plasma of cognitively impaired MS patients, compared to cognitively preserved MS patients [110]. let-7b-5p targets a variety of transcripts of synaptic genes [120] and downregulates the expression of several inflammatory genes, thus having an impact on immune cell function [121]. All these data highlight the need for rigorous studies that may move potential candidates to new MS biomarkers.

## 5. EV Is MS Therapy

EVs are considered promising candidates for therapeutic purposes thanks to their ability to carry molecules and to their specific characteristics. The release of EVs can be inhibited to obtain anti-inflammatory or neuroprotective effects [122,123]. EVs can be used as therapeutic agents by themself taking advantage of their stability, good biocompatibility, high targeting specificity, and low immunogenicity [124]; in addition, they are able to cross blood vessels and the BBB [125]. EVs can also be engineered to increase their efficiency as therapeutics: EVs can be loaded with specific molecules and drugs, and they can also be modified in their surface ligands to obtain receptor-mediated tissue targeting [126,127].

Therapeutic applications of EVs are still under investigation in MS, with several studies being conducted on pre-clinical models, with a particular focus on MS inflammation and EVs' potentiality for microglia activation.

### 5.1. EVs Impact on Immune Cells

EVs from DCs expressing the membrane-associated TGF-β1 intravenously injected in EAE mice after disease onset can inhibit the development and progression of the disorder [128]. The treatment impaired Ag-specific Th1 and IL-17 responses and reduced the levels of Th17 cells. Moreover, the EV treatment allowed Treg cells to maintain their regulatory capacity of Treg cells. These results suggest that EVs may be a powerful immunosuppressive tool to effectively inhibit the development and progression of EAE [128]. EVs from adipose stem cells (ASCs) may have an immunosuppressive effect on EAE. The preventive intravenous injection of EVs from ASC before EAE onset significantly decreases the severity of the disorder [129]. In addition, the spinal cord inflammation and demyelination were reduced. Indeed, in vitro ASC-derived EVs inhibited integrin-dependent adhesion of encephalitogenic T-cells, and such cells display a significantly decreased rolling and adhesion in inflamed spinal cord vessels, thus suggesting that ASC-derived EVs ameliorate EAE pathogenesis primarily by the inhibiting T-cell invasion of the CNS. However, the therapeutic treatment with derived EVs did not ameliorate established EAE. On the other hand, a 2020 study showed that, after the induction of EAE in mice, the intravenous injection of EVs from human adipose-derived mesenchymal stem cells (ADSC-EVs) reduced the clinical score [124]. In the EVs-treated mice, myelin oligodendrocyte glycoprotein-induced proliferation of splenocytes was significantly lower, compared to control mice, and the percentage of demyelination areas significantly decreased, suggesting that hADSC-EVs can attenuate EAE through diminishing proliferative potency of T-cells, leukocyte infiltration, and demyelination [130]. Furthermore, the intranasal administration of EVs from mesenchymal stem cells (MSCs) has been tested on the EAE mouse model after immunization [131]. Treated mice showed a significant decrease in clinical scores, and the MSC-derived EVs treatment more effectively alleviated the clinical scores and histological lesions of the CNS, compared to the treatment with MSCs. The reduction of clinical symptoms was associated with higher immunomodulatory responses [131]. Moreover, the intravenous administration of EVs release from MSCs stimulated by IFNγ (IFNγ-EVs) may reduce the mean clinical score of EAE mice, reduce demyelination, decrease neuroinflammation, and increase the number of Tregs within the spinal cords of such mice [132]. RNA sequencing suggested that IFNγ-EVs contain anti-inflammatory RNAs and multiple anti-inflammatory and neuroprotective proteins [132]. Giunti and colleagues identified nine miRNAs significantly dysregulated in IFN-γ-primed MSCs. However, these microRNAs were present at different levels in the MSC-derived EVs [133]. In particular, the authors showed that miR-467f and miR-466q modulated the pro-inflammatory phenotype of activated microglia in vitro and administered the EVs to EAE mice. The EVs decreased the expression of neuroinflammation markers in the spinal cord of treated mice; however, they did not have an impact on the disease course [133]. A recent study evaluated the anti-inflammatory effect of MSCs and MSC-derived EVs against the proliferation of T-cells isolated from MS patients and healthy subjects in vitro [134]. Both MSCs and MSC-derived EVs reduced the proliferation of conventional T-cells producing IFN-γ and IL-17, whereas they increased

the IL-10-producing cells. Interestingly, MSC-derived EVs had higher immune-modulating properties on conventional T-cell responses, compared to MSCs, supporting the use of EVs in potential MS therapies [134]. Human umbilical cord blood (UCB) cell-derived EVs may play immunosuppressive roles; a 2020 study examined the immunosuppressive potential of UCB plasma-derived extracellular vesicles (CBP EVs), compared to that of adult blood plasma-derived extracellular vesicles (ABP EVs), finding that in vitro CBP EVs contain more immunosuppression-related proteins than ABP EVs and their suppression of T-cell proliferation was due to apoptosis and cell cycle arrest [135]. The immunosuppressive pro-prieties were also confirmed in vivo: the clinical scores of CBP EV-treated EAE mice were significantly lower than those of the group treated with ABP EV. CBP EVs can inhibit T-cell proliferation and induce Treg and myeloid-derived suppressor cell (MDSCs) differentiation, probably acting on the inhibition of IL-2 signaling [135].

EVs may also be used to induce immune tolerance for a specific self-antigen. Several approaches for antigen-specific suppression of autoimmune neuroinflammation have been proven effective in EAE, but the use of such strategies for MS patients has been hampered by the uncertainty of the best myelin antigens in human treatments [136–138]. Casella and colleagues developed a therapeutic strategy based on oligodendrocyte (Ol)-derived EVs (Ol-EVs), naturally carrying multiple myelin antigens [139]. The intravenous injection of Ol-EVs in mice with EAE led to a reduction of disease pathophysiology in a myelin antigen-dependent manner, both prophylactically and therapeutically. Ol-EVs restored immune tolerance by inducing immunosuppressive monocytes and promoting apoptosis of autoreactive CD4$^+$ T-cells. In addition, the study showed that also human oligodendrocytes released EVs carrying relevant myelin antigens, thus reinforcing their putative use in MS therapy [139].

*5.2. EVs Impact on Microglia*

Bone marrow mesenchymal stem cells (BMSC)-derived EVs may have a therapeutic potential in MS, due to the bioactive compounds that they are carrying. In EAE rats, the tail vein injection of BMSC-derived EVs led to a significant reduction of EAE severity, reduced infiltration of inflammatory cells into the CNS, and decreased demyelination, compared to untreated EAE animals [140]. The study showed increased levels of M2-related cytokines, such as interleukin (IL)-10 and transforming growth factor (TGF)-β, whereas M1-related tumor necrosis factor (TNF)-α and IL-12 levels decreased significantly. Moreover, the EVs-treated rats displayed significantly increased protein and mRNA expression levels of M2 phenotype markers, whereas M1 marker expression decreased, further suggesting that BMSC-EVs may attenuate inflammation and demyelination by regulating the polarization of microglia [140]. BMSC-derived EVs containing miR-367–3p may reduce microglial ferroptosis both in vitro and in vivo [141]. EAE mice intrathecally injected with BMSC-derived EVs show a reduction in the severity of EAE thanks to the suppression of the ferroptosis [141]. Furthermore, BMSC-derived EVs containing miR-23b-3p, injected into EAE mice, were able to reduce microglial pyroptosis by decreasing microglial inflammation, and finally alleviating the severity of EAE [142]. Laso-García and colleagues analyzed the effect of the intravenous injection of EVs released from MSCs from human adipose tissue on Theiler's murine encephalomyelitis virus (TMEV)-induced demyelinating disease, a progressive model of MS: the EV administration improved motor deficits, decreased brain atrophy, increased cell proliferation in the subventricular zone, promoted remyelination, and reduced inflammatory infiltrates in the spinal cord. Moreover, the treatment with EVs can modulate neuroinflammation and the activation state of microglia [143].

*5.3. EVs Impact on Oligodendrocytes and Remyelination*

In 2014, Pusic and colleagues showed that EVs from DCs stimulated with low-level IFNγ contained microRNAs which, in vitro, can increase baseline myelination, decrease oxidative stress and promote remyelination following acute lysolecithin-induced demyelination [144]. Such EVs, when nasally administered to rats, lead to increased CNS myelina-

tion, thus suggesting their potential impact on oligodendrocytes and their potential use as therapeutics to promote remyelination [144]. A more recent study analyzed MSC-derived EVs from rhesus monkey MSCs which were intravenously injected into the EAE model: MSC-EVs can cross the BBB and target neural cells, and significantly improve neurological outcome [145]. In the treated animals, an increase in the numbers of both newly generated oligodendrocytes and mature oligodendrocytes was detected. An increase in the level of myelin basic protein (MBP) was noticed together with a decrease in neuroinflamma-tion, due to the increase of the M2 and decrease in the M1 microglia phenotype and the inhibition of the TLR2/IRAK1/NFκB pathway. This study provides new insight into the possibility of using EVs for remyelination and modulation of neuroinflammation in the CNS [145]. EVs from placenta MSCs (PMSCs) display a good potentiality as a therapeutic tool for MS. Indeed, EAE mice treated with high-dose PMSC-EVs showed amelioration in motor function, decreased DNA damage in oligodendrocytes and increased myelination within the spinal cord, as well as mice injected with PMSCs [146]. In addition, in vitro analyses revealed that PMSC-derived EVs promote myelin regeneration by inducing the differentiation of OPCs into mature myelinating oligodendrocytes, thus supporting the hypothesis that the PMSCs' mechanism of action is mediated by the secretion of EVs [146].

*5.4. EVs as Vectors*

EVs can act as drug delivery vectors if opportunely modified. In a 2019 study, the carboxylic acid-functionalized LJM-3064 aptamer was covalently conjugated to the amine groups on the EV surface [147]. LJM-3064 is a DNA aptamer, smaller and more robust than IgM monoclonal antibodies, with a considerable affinity toward myelin and the potential ability to promote the remyelination process in mice [148]. In vitro, the aptamer-exosome bioconjugate may promote the proliferation of oligodendroglia cell line (OLN93) and, when administered in vivo in female C57BL/6 mice as a prophylactic measure, it robustly suppressed the inflammatory response by reducing Th1 response and increasing Treg population, lowered demyelination lesions in the CNS, and reduced the severity of EAE [147]. A recent paper aimed to reduce the inflammatory response in the CNS and peripheral system in a mouse model of MS by using EVs derived from macrophages, which were shown to be strongly co-localized with microglia loaded with resveratrol [149]. The intranasal administration of such EVs to the mouse model significantly inhibited inflammatory responses and effectively improved the clinical evolution of MS in vivo.

EVs from neural stem cells (NSCs) modified to express the ligand of the platelet-derived growth factor (PDGF-A) can achieve local delivery [150]. PDGF is a growth factor expressed by neurons, astrocytes, monocytes/macrophages, NK, and T-cells [151]. PDGF plays a critical role in OPC proliferation, development, and migration, by targeting PDGFRα, a receptor expressed by immature OPCs [152,153]. EVs expressing PDGF-A have a greater capability to target oligodendrocytes lineage [150]. In addition, Xiao and colleagues loaded such EVs with triiodothyronine (T3), a thyroid hormone that promotes oligodendrocyte development and survival, and is involved in remyelination [154–156]. An excessive systemic administration of T3 may have serious side effects; therefore, the loading of EVs with T3 is a more promising approach for CNS drug delivery [150,157]. Indeed, engineered EVs expressing PDGF-A and loaded with T3 led to significant suppression of disease development in mice with EAE, compared to those treated with EVs expressing PDGF-A or T3 alone, allowing remyelination in lesions, promoting OPC differentiation, and protecting oligodendrocytes [150].

As mentioned before, the re-myelination of myelin sheaths in MS therapy can be induced by the differentiation of OPCs, located at the myelin injury region, into mature oligodendrocytes to form new myelin sheaths. In this regard, brain-derived neurotrophic factor (BDNF) may be promising molecules, playing a key role in the regulation of OPC proliferation and differentiation into mature oligodendrocytes [158]. The intranasal admin-istration of brain-targeted engineered EVs loaded with BDNF to demyelination model mice effectively routed BDNF to the brain and had a significant effect on remyelination [159].

Moreover, it improved the motor coordination ability of the treated mice, providing a strategy for efficient drug delivery [159]. MicroRNAs have been suggested as likely tools to induce remyelination by promoting the differentiation of oligodendrocyte precursor cells. Osorio-Querejeta and colleagues showed that engineered EVs overexpressing miR-219a-5p induce OPC differentiation in vitro and, when intranasally administered to mice with EAE, such EVs lead to successfully decreased clinical scores [160]. Interestingly, this work showed that EVs carrying miR-219a-5p have the highest oligodendrocyte precursor and more effectively cross the BBB, compared to poly lactic-co-glycolic acid (PLGA) nanoparticles and liposomes, two synthetic and clinically approved drug delivery vectors, making them the most promising microRNA delivery system [160]. A recent work focused on engineered EVs as a trigger of re-myelination [161]. The authors isolated the EVs from mice neural stem cells and modified them to express the PDGFRα ligand on the membrane, target OPCs in the lesion area, and carry bryostatin-1 (Bryo). Cuprizone-fed mice, a model used to investigate demyelination and remyelination processes [162], were treated; EVs with Bryo showed a powerful therapeutic effect, compared with Bryo alone, being able to significantly improve the protection of myelin sheaths and promoting remyelination. Furthermore, a significant block of astrogliosis and axon damage, and inhibited pro-inflammatory microglia were detected [161].

## 6. Conclusions

In this review, we summarized the current knowledge on the involvement of EVs in the pathogenesis of MS and on the possible use of such carriers as diagnostic and therapeutic agents, also focusing on the role of EVs in the pathogenic mechanism underlying MS and EAE. EVs released from a plethora of cells, including myeloid cells, leukocytes, BBB-endothelial cells, microglia, and astrocytes, are involved in the complex pathogenesis of this disorder, contributing to BBB damage, spreading pro-inflammatory signals, and altering CNS functioning. Although with differences in study designs and technical aspects, literature data suggest that MS EVs act as inflammatory mediators by inducing Th17 cell differentiation and maturation, and by inhibiting the differentiation of functional Treg cells. In addition, they are involved in the disruption of the BBB, and contribute to the infiltration of immune cells into the CNS. In the CNS, EVs are released from neurons and glial cells, and studies on MS focus mainly on microglia-derived EVs: they may contribute to the formation of focal inflammatory lesions, due to their ability to mediate inflammation, and may alter synaptic plasticity. However, we are still far from a complete understanding of the role of EVs in MS, and future studies will be important to completely figure out the functional effects of EVs and their specific modes of action in the several types of MS. It is important also to consider that cell-specific EVs can be studied in vitro in a relatively easy way, methodological difficulties hinder the studies on the EV biology in vivo. For years a proper standardization of EVs isolation and characterization was lacking, but researchers are working to overcome this issue by improving the right isolation method, based on the nature of biofluids containing EVs. Standardized protocols should be developed to promote harmonization among laboratories and to allow tighter control for EV characterization.

This consideration must be taken into account also when considering the potential use of EVs as biomarkers. At present, proteins and miRNAs contained in EVs are among the most analyzed molecules in order to find biomarkers for the early diagnosis of MS, but also for distinguishing MS subtypes and checking the effect of therapy. However, due to the lack of standard protocols, their validation for the use of EVs as biomarkers for diagnosis in a clinical setting is crucial. In addition, the application in the clinical practice must also consider the pre-analytical phase: several variables and factors can affect EV isolation and recovery, including the type of body fluid used for EV isolation. For example, plasma and serum are rich in EV content, but are also enriched in lipoproteins which can be isolated together with the EVs. The isolation step must be carefully planned, and the right techniques must be chosen to avoid this co-isolation. On the other hand, CSF has fewer contaminants, but contains fewer EVs; thus, the EV concentration is also crucial.

EVs display great potential in diagnosis for the treatment of MS. Pre-clinical studies on EAE models focused the attention on using EVs to inhibit inflammatory processes, for example, by reducing the levels of Th17 cells, inducing Treg cells to maintain their regulatory capacity of Treg cells, and reducing leukocyte infiltration. Other studies investigated the use of EVs to attenuate demyelination by microglial activation, or even inducing remyelination, thanks to their effect on oligodendrocytes. In addition, EVs can act as drug delivery vectors, which can cross the BBB. Although significant advances have been conducted in the study of EVs in MS therapy, EV use still has some limitations arising from the knowledge gaps: many aspects of EVs biology, biodistribution, and cell-targeting properties remain to be elucidated. Thus, further research efforts must be performed to encourage further research for the use of EVs as MS treatment.

**Author Contributions:** Conceptualization, C.P. and C.O.; methodology, C.P.; resources, C.P.; data curation, C.P.; writing—original draft preparation, C.P.; writing—review and editing, C.P., C.O., M.C. and R.B.; visualization, C.P. and C.O.; supervision, C.P. and C.O.; funding acquisition, C.P., C.O., M.C. and R.B. All authors have read and agreed to the published version of the manuscript.

**Funding:** This research was supported by the Dipartimenti di Eccellenza Program (2018–2022)—Ministry of Resesearch—To the Department of Biology and Biotechnology "L. Spallanzani" University of Pavia.

**Conflicts of Interest:** Roberto Bergamaschi has served on scientific advisory boards for Biogen, Merck-Serono, Novartis, Sanofi-Genzyme, has had travel and congress expenses sustained by Biogen, Bristol Myers Squibb, Janssen, Merck-Serono, Roche, Sanofi-Genzyme, received honoraria for speaking engagement from Biogen, Bristol Myers Squibb, Janssen, Merck-Serono, Novartis, Sanofi-Genzyme, received research support from Biogen, Merck-Serono, Novartis, Sanofi-Genzyme.

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
