# Peer review of "Roles of Extracellular Vesicles in Multiple Sclerosis: From Pathogenesis to Potential Tools as Biomarkers and Therapeutics"

_2813-3064, doi:10.3390/sclerosis1020011_

Round 1

Reviewer 1 Report

The subject of the role of EVs in the mechanism of MS is well-addressed in your review. While your proposals of their potential as biomarkers and therapeutics is based on appropriate studies and literature (with adequate extrapolation of your part), the reader is left with questions about the large data presented in the tables and Figure 1.

1) The discussion of the tables data does not provide information on which modality would be a more realistic and practical biomarker. The hypothetical basis is not questioned, however in real-world MS care, a simple and non-expensive test it's desirable, hopefully easy to be determined, preferably from blood, etc. Your comment will be appreciated.  

2. Figure 1 is not clear. The representation of the EVs in the illustration is not clearly explained. The figure also indicates an immune flow across the BBB from the CNS to peripheral blood which is not properly discussed in the text. 

Adequate.

Author Response

1) We added a column to both Table 1 and 2 “Potential application” to indicate the potential application of the mentioned biomarker. In addition, we decided to add paragraph 4.3 Clinical applications of potential EV-derived biomarkers to better explain the potentiality and the pitfalls of EVs as biomarkers for MS. As these studies are still in their early stages, the biomarkers reported in the tables need to be confirmed in larger cohorts, in the next future. We highlighted that, at moment, the studies on EVs biomarkers for MS are pretty heterogeneous and lack solid standardization. In addition, we now focus on the promising field of EV-derived miRNA profiling, drawing the attention on the microRNAs reported in the tables that were described in more than one paper.   

2) Thanks to referee’s suggestion, we now modified the figure to make clear that the arrows indicate a potential EVs flux across the BBB. Indeed, EVs have been described to be able to cross the BBB in both direction (see for example Banks WA et al.,Transport of Extracellular Vesicles across the Blood-Brain Barrier: Brain Pharmacokinetics and Effects of Inflammation, 2020), but other studies are needed to confirm the mechanisms and the implications. 

Reviewer 2 Report

This is a very well written review on a hot topic with great potential. Current knowledge on the subject is systematically and thoroughly presented and the potential use of EVs as biomarkers or therapeutics is discussed. 

Please note the following  typos or sentences requiring clarification/rephrasing:

line 358: ..thanks to their ability...

line 428: based on

line 463.. and promote remyelination....

line 558-560: please rephrase

Author Response

We thank the reviewer for pointing out these typos and the necessity to rephrase some sentences. The typos have been corrected and the indicated sentences rephrased.